# ICPC-Eval: Probing the Frontiers of LLM Reasoning with Competitive Programming Contests

**Shiyi Xu**[1,2,3,*] **Yiwen Hu**[1,*]**, Yingqian Min**[1]**, Zhipeng Chen**[1]**,**
**Wayne Xin Zhao**[1,2,3,†]**, Ji-Rong Wen**[1,2,3]

[1] Gaoling School of Artificial Intelligence, Renmin University of China
[2] Beijing Key Laboratory of Research on Large Models and Intelligent Governance
[3] Engineering Research Center of Next-Generation Intelligent Search and Recommendation, MOE
{shiyixu45, batmanfly}@gmail.com

## Abstract

With the significant progress of large reasoning models in complex coding and reasoning tasks, existing benchmarks, like LiveCodeBench and CodeElo, are insufficient to evaluate the coding capabilities of large language models (LLMs) in real competition environments. Moreover, current evaluation metrics such as Pass@K fail to capture the reflective abilities of reasoning models. To address these challenges, we propose **ICPC-Eval**, a top-level competitive coding benchmark designed to probing the frontiers of LLM reasoning. ICPC-Eval includes 118 carefully curated problems from 11 recent ICPC contests held in various regions of the world, offering three key contributions: 1) A challenging realistic ICPC competition scenario, featuring a problem type and difficulty distribution consistent with actual contests. 2) A robust test case generation method and a corresponding local evaluation toolkit, enabling efficient and accurate local evaluation. 3) An effective test-time scaling evaluation metric, Refine@K, which allows iterative repair of solutions based on execution feedback. The results underscore the significant challenge in evaluating complex reasoning abilities: top-tier reasoning models like DeepSeek-R1 often rely on multi-turn code feedback to fully unlock their in-context reasoning potential when compared to non-reasoning counterparts. Furthermore, despite recent advancements in code generation, these models still lag behind top-performing human teams. We release the benchmark at: https://github.com/RUCAIBox/Slow_Thinking_with_LLMs

## 1 Introduction

Large language models (LLMs) have demonstrated exceptional performance across a diverse range of tasks [1]. Recent advancements in reasoning-focused models, such as OpenAI's o1/o3 series models [2], DeepSeek-R1 [3], and Gemini 2.5 Pro Exp [4] have significantly advanced their problem analysis and reasoning capabilities. Consequently, competitive programming problems, which necessitate the translation of complex mathematical logic into executable code, are widely employed for such evaluations [5–7]. Moreover, problems in real competitions usually involve understanding the meaning of problem statement, making competitive programming problems a comprehensive test of an LLM's intelligence.

---

[*]Equal Contributions.
[†]Corresponding author.

39th Conference on Neural Information Processing Systems (NeurIPS 2025) Track on Datasets and Benchmarks.

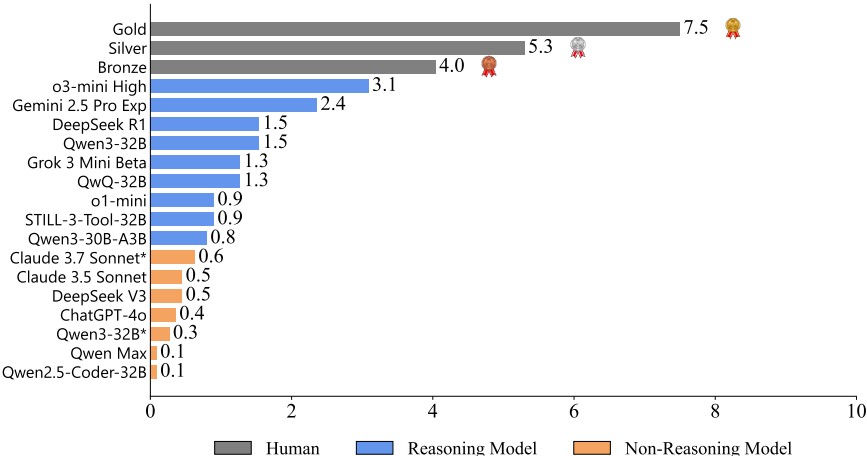

Figure 1: Average number of problems solved per contest (typically 12 problems) by AI models compared to human ICPC medalists. Despite their strong reasoning capabilities, current top models are still unable to achieve medal-winning performance in ICPC competitions.

However, existing programming benchmarks still face two main challenges. **Firstly, they are of relatively low difficulty**. With the rapid advancement of large language models (LLMs), these models have achieved high scores on current benchmarks. For example, many LLMs pre-trained on code data can score over 98% percentile on code completion benchmarks, such as rating in CodeElo [7]. Similarly, the problems from from active coding platform like LiveCodeBench [5] and USACO [6] do not reach the top levels of algorithmic competition difficulty, making them increasingly solvable by powerful reasoning models. This trend diminishes the benchmarks' discriminative power. **Secondly, the evaluation methodology lacks accessibility and realism**. While most difficult problems from actual competitions offer online evaluation on various Online Judges such as Codeforces, LeetCode, AtCoder, and Luogu, their private test cases are typically not publicly disclosed. Consequently, benchmarks like CodeElo[7], and LeetCode-Hard[8] rely on direct submission to these platforms, creating barriers for researchers seeking to evaluate their own models. Moreover, the widely used Pass@K metric fails to capture the iterative refinement process inherent in authentic problem-solving, where even top-tier competitors rarely produce correct solutions on their first attempt. Additionally, real competition scenarios provide concise feedback on submissions, such as timeouts and incorrect answers, which are not reflected in the Pass@K metric. This limitation further undermines its relevance in evaluating model performance in realistic contexts.

To address these challenges, we propose **ICPC-Eval**, a top-level competitive coding benchmark designed to evaluate the advanced reasoning capabilities of LLMs. Our goal is to comprehensively tackle issues related to problem difficulty, special judges, local evaluation, and suitable metrics for assessing reasoning models. To achieve this, we collect sufficiently challenging problems from International Collegiate Programming Contest (ICPC) contests, which are prestigious competitions for university students. Specifically, we gather problems from 11 ICPC contests hosted on the QOJ and Vjudge [3] platforms. Next, we eliminate problems that contain essential non-textual images, interactive elements, or lack a standard solution, ensuring that the remaining problems can be solved and verified well. Ultimately, we retain a total of 118 problems. Among them, we develop SPJs for 12 problems that involved floating-point output or multiple valid solutions, striving to closely replicate the problem types and difficulty distribution of actual competitions. We also tag these problems with type labels. These problems represent the most challenging programming competitions and are sufficient to pose significant challenges to current state-of-the-art reasoning models.

To address the challenges of inaccessible private test cases and the over-reliance on Online Judges, ICPC-Eval introduces a robust test case generation method. This process utilizes large language models (LLMs) to synthesize C++ input data "generators" for each problem. These generators are specifically prompted to create both random inputs (sampled uniformly from defined ranges) and

---

[3] https://qoj.ac/ and https://vjudge.net/

Table 1: Comparison of different programming evaluation benchmarks. Each benchmark is categorized by its source, difficulty level, locality (*i.e.* whether can be evaluated locally), special judge support (SPJ), whether it ensures zero false positives (Zero FP), and the evaluation metric.

| Name | Source | Difficulty | Local? | SPJ? | Zero FP? | Metric |
|------|--------|:----------:|:------:|:----:|:--------:|--------|
| HumanEval | Handwritten | ★ | ✔ | ✗ | ✗ | Pass@K |
| USACO | USACO | ★★ | ✔ | ✗ | ✔ | Pass@K |
| LiveCodeBench | LeetCode, *etc*. | ★★ | ✔ | ✗ | ✗ | Pass@K |
| CodeElo | CodeForces | ★★ | ✗ | ✔ | ✔ | Pass@K |
| ProBench | ICPC | ★★★ | ✗ | ✔ | ✔ | Pass@K |
| ICPC-Eval | ICPC | ★★★ | ✔ | ✔ | ✔ | Refine@K |

challenging corner-case inputs (based on edge cases and specially structured instances identified from the problem statement). Outputs for these generated inputs are produced using known accepted solutions, and the entire set of synthesized test cases is rigorously validated to ensure they correctly identify errors in a curated collection of known incorrect programs (*e.g.* those that fail with `Wrong Answer` or `Time Limit Exceeded` on the online judge). This approach creates an efficient and accurate local evaluation toolkit. Furthermore, to capture the critical iterative refinement process involved in solving complex competitive programming problems, we simulate the actual competition environment and propose Refine@K as an effective test-time scaling evaluation metric. This metric assesses an LLM's ability to improve its solution within a budget of K attempts. After the initial code generation based on the problem, if the solution fails to compile or passes example cases but fails hidden test cases, the model receives specific execution feedback and is prompted to iteratively refine its code within the K-attempt limit. This approach provides a more nuanced evaluation of a model's reasoning capabilities compared to traditional simple sampling metrics (*e.g.* Pass@K).

We comprehensively evaluate 15 state-of-the-art LLMs, with the results shown in Figure1 and Table3. We observe that even the best-performing model such as o3-mini High still exhibits a significant performance gap compared to top human participants, highlighting the high difficulty level of ICPC-Eval. Additionally, we find that Refine@K scales robustly with increasing output lengths across models, indicating its promise as an efficient method for evaluating test-time scaling. Furthermore, through ablation studies, we verify that Refine@K is more suitable than Pass@K for evaluating the reasoning capabilities of models.

The main contributions of our work can be summarized into three aspects as follows.

• **A challenging benchmark** featuring top-difficulty problems curated from recent ICPC, ensuring a rigorous test of advanced reasoning without data collaboration.

• **A novel test case generation and validation methodology** that leverages LLMs to create comprehensive local test suites, including a local evaluation toolkit, enabling robust and accessible offline assessment.

• **An effective test-time scaling evaluation metric, Refine@K**, designed to measure an LLM's ability to iteratively refine its solutions based on execution feedback over multiple attempts.

## 2  Related Work

**Code Benchmarks.**  Early benchmarks like HumanEval [9] and MBPP [10] focus on relatively simple, manually curated function generation tasks. However, these benchmarks face limitations in scalability and comprehensive test coverage. To assess more complex reasoning capabilities, APPS [11] and CodeContests [12] introduce problems from competitive programming. xCodeEval [13] further expands the scope by incorporating multilingual and multitask programming challenges. While these benchmarks may include a limited set of local test cases, their verification primarily relies on publicly available problem descriptions and sample test cases. This reliance is often inadequate for rigorous solution verification, as crucial hidden test cases remain undisclosed. More recent efforts, such as LiveCodeBench [5] and USACO [6], curate tasks from active coding platforms. While these benchmarks increase task difficulty, they may not consistently reflect the highest levels of al-

Table 2: Distribution of contest problems across algorithmic tags. Each problem may be associated with one or more tags. 'WFs' and 'CFs' denote World Finals and Continental Finals, respectively.

| Domain | Topic | Count | | |
|---|---|---|---|---|
| | | WFs & CFs | Regionals | Total |
| Algorithm Basics | Greedy, Divide-and-conquer, *etc.* | 7 | 27 | 34 |
| Computational Geometry | Sweep Line, Rotating Calipers, *etc.* | 6 | 11 | 17 |
| Data Structure | Segment Tree, Binary Search Tree, *etc.* | 6 | 24 | 30 |
| Dynamic Programming | Knapsack, DP on Trees, Bitmask, *etc.* | 11 | 27 | 38 |
| Graph Theory | Dijkstra, Network Flow, *etc.* | 4 | 22 | 26 |
| Mathematics | Combinatorics, Number Theory, *etc.* | 15 | 33 | 48 |
| Search Algorithm | DFS, BFS, Backtracking, *etc.* | 15 | 20 | 35 |
| String Algorithm | KMP, Z-algorithm, Suffix Array, *etc.* | 5 | 1 | 6 |
| All | | 31 | 87 | 118 |

gorithmic complexity found in ICPC contests. Additionally, they often lack support for local special judges (SPJs) and do not ensure complete test coverage, which can lead to false positives. Benchmarks like CodeElo [7], which focus on high-difficulty problems from Codeforces, typically require submissions to online judges. This requirement limits local reproducibility and restricts access to SPJ-based evaluations. These limitations underscore the need for benchmarks that combine extreme difficulty with robust, comprehensive, and fully accessible local evaluation infrastructures.

**Iterative Refinement.** Iterative refinement is an intuitive approach that enhances model performance by incorporating execution feedback in a multi-turn dialogue setting. Previous studies have primarily focused on training-based methods to elicit the models ability for self-reflection [14, 15]. Some work has also explored incorporating execution feedback directly into inference, but such methods tend to underperform compared to multiple sampling when applied to non-reasoning models [16, 17]. With the recent advances in reasoning models, reflective thinking has emerged even without explicit feedback [2, 3]. Our goal is not to propose a new method for improving model capabilities, but rather to introduce an evaluation metric that models the process of reflectionbetter aligning with real-world usage scenarios.

## 3 ICPC-Eval: Task and Construction

In this section, we describe the ICPC-Eval benchmark in detail, including its problem collection, data distribution, test case generation, and evaluation metric design (*i.e.* Refine@K). We present a basic comparison of ICPC-Eval and other coding benchmarks in Table 1.

### 3.1 Problem Collection

We curate a total of 11 ICPC contests, comprising 139 raw problems. Among these, 3 are from ICPC World Finals or Continent Finals, and 8 come from Regional contests. Our selection process follows these criteria: 1) **recency**: We prioritize contests held from October to December 2024, except for the 2023 ICPC World Finals, as their publicly available manuals are limited. We can update our benchmark annually using the latest ICPC problems, which helps minimize the risk of data contamination. 2) **minimal contamination**: We verify online that platforms like VJudge display user-submitted solutions as images and employ strict anti-crawling mechanisms, reducing the likelihood of these contests being included in the training corpus of models. 3) **representativeness**: We ensure the contests are representative of the typical problem distribution and difficulty found in ICPC contests. This approach ensures that our curated contests are both current and reflective of the ICPC's standards, while also minimizing the risk of data contamination.

To ensure the quality and consistency of the dataset, we implement a series of filtering and formatting steps on the collected problems. Initially, we eliminate 8 problems that fall into the following categories: 1) **non-textual images**, such as diagrams or pictures, 2) **interactive problems**, or 3)

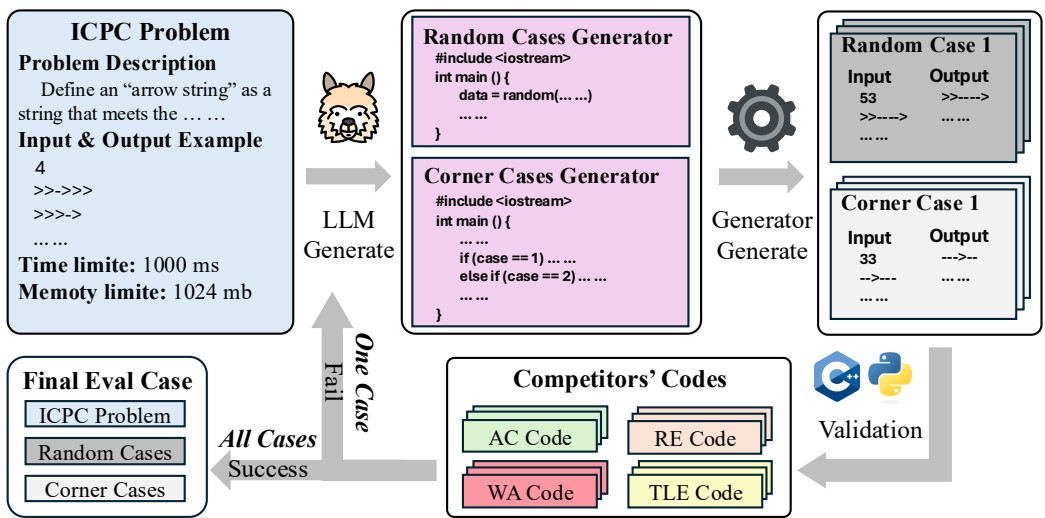

Figure 2: The complete pipeline for test case generation and validation, enabling efficient local evaluation.

**lacking a standard solution**. However, we retain problems that include tables or textual images, provided they can be accurately represented in plain text without losing information. Additionally, out of the 25 problems that utilize special judges, we exclude 13 problems for which it was not feasible to develop correct and efficient special judges. We retain the remaining 12 problems and write dedicated special judges for each of them. Finally, we standardize all remaining problem descriptions into a unified LATEX format to facilitate better comprehension and processing by models. After these data cleaning steps, we obtain a total of 118 problems, which constitute the final ICPC-Eval test set.

## 3.2 Problem Distribution

Due to the high difficulty of ICPC problems, a single problem may involve one or more algorithmic domains. Therefore, instead of assigning each problem to a mutually exclusive category, we annotate each problem with its relevant algorithmic tags. Based on the common types of recent ICPC problems, we divide the algorithmic domains into eight areas: *Algorithm Basics* (*Algo*), *Dynamic Programming* (*DP*), *Mathematics* (*Math*), *Data Structure* (*DS*), *Graph Theory* (*GT*), *Computational Geometry* (*CG*), *Search Algorithm* (*SA*), and *String Algorithm* (*Str*). To annotate these tags, we utilize the `gemini-2.5-flash-preview-04-17-thinking` model, providing it with detailed classification criteria, problem statements, and their correct solution code. By reviewing the AI-generated solution explanations and classification suggestions, we manually assign tags to each problem. The detailed classification criteria and prompt are available in Appendix B.2.

We present the distribution of problems across these tags in Table 2. As illustrated, the majority of problems involve at least one advanced algorithmic domain: *Mathematics*, *Dynamic Programming*, or *Search Algorithm*. Additionally, we observe that the World Finals and Continent Finals provide a more comprehensive examination of algorithmic knowledge. This indicates that ICPC-Eval establishes a notably more challenging baseline for state-of-the-art models.

## 3.3 Test Case Generation

During data collection, we observe that the lack of robust local test cases often poses significant inconvenience to evaluation. Existing code evaluation benchmarks either necessitate the use of crawlers for OJ submissions or involve problems that are overly simplistic. To tackle the difficulties, we propose a test case data construction process leveraging LLMs to generate robust local test cases.

**Input Generator.** We utilize the `gemini-2.5-pro-preview-03-25` API to synthesize input data generators written in C++. These generators are designed to produce input data tailored to specific problems. For each problem, two types of generators are implemented: **a random genera-**

**tor**, $G_{rand}$, which samples uniformly from the defined data range, and **a corner case generator**, $G_{corner}$, which generates inputs based on carefully crafted edge cases. Comprehensive prompts and examples for test case generators are provided in Appendix B.3.

**Output Generation and Validation.** To generate outputs for each input data point, we collect one `Accepted` program for each problem from QOJ. The correctness of these programs is rigorously validated using a feature called "Hacks" on QOJ, where the community contributes additional test cases, beyond the official ones, to identify potential flaws in the programs. To further validate the reliability of the synthesized test cases, we manually collect three programs with statuses of either `Wrong Answer`, `Time Limit Exceeded`, or `Runtime Error`. We compare the evaluation results on these curated programs to validate these test cases are exhibiting similar behaviour with official test cases. We ask the model to regenerate the generators that failed in any these check to ensuring zero false positives of test cases. As shown in Figure 2, our generated test cases have successfully differentiate the correct and incorrect programs.

### 3.4 Refine@K: Towards Better Test-time Scaling Metrics

To accurately evaluate the correctness of code generation, the Pass@K metric has been proposed in previous research [18]. We begin by reviewing the commonly used Pass@K evaluation method. Pass@K aims to estimate the probability that, given a sampling budget of K, at least one of the generated samples will pass the test cases on average. To better approximate this expected probability, LLMs typically sample $N$ code completions (where $N \geq K$), and compute the metric based on the accuracy of each sample as follows:

$$\text{Pass@K} := \mathbb{E}_{\text{Problems}} \left[ 1 - \frac{\binom{N-C}{K}}{\binom{N}{K}} \right]$$

where $C$ is the number of samples that pass all unit tests. Pass@K is a commonly used metric to assess the performance of programming at test-time scaling by gradually increasing $K$ [9, 19]. However, with the development of reflection and reasoning abilities in recent LLMs, it underestimates the comprehensive capabilities of these models. This is because in real-world chat scenarios, these models are often used in multi-turn conversations with environmental feedback instead of sampling $N$ responses from an i.i.d. distribution. This challenge is particularly pronounced in ICPC-style competitions, where models are often faced with problems of high cognitive complexity, for which human competitors similarly rely on multiple submission attempts to reach solutions. In fact, according to official statistics from the 2024 ICPC Asia Chengdu Regional Contest, teams submit an average of 1.95 attempts per solved problem, underscoring the centrality of feedback-driven refinement in realistic competition settings. Therefore, to more accurately assess the algorithmic reasoning capabilities of models, we propose a new metric, **Refine@K**, *i.e.* whether the model can pass the test within $K$ response and refinement chances:

$$Response_i = \begin{cases} \text{LLM}(Problem) & \text{if } i = 1, \\ \text{LLM}(Problem, Response_{i-1}, Feedback_{i-1}) & \text{if } 1 < i \leq K. \end{cases}$$

It measures a model's true algorithmic capability when provided with additional external information. In the first turn, the model receives the full problem statement in LaTeX format, including the task description, input/output specifications, and example test cases. In subsequent turns, the model is additionally provided with its previous response and corresponding evaluation feedback. We provide detailed information about how feedback is incorporated during evaluation in Section 4.1. We also demonstrate in Section 5.2 that Refine@K serves as a more effective test-time estimator than Pass@K when evaluating reasoning models.

## 4 Experiment

### 4.1 Experiment Setup

**Models.** We comprehensively evaluate 15 state-of-the-art LLMs. Unless otherwise specified (via API endpoint), all evaluations are conducted using open-weight models hosted with vLLM

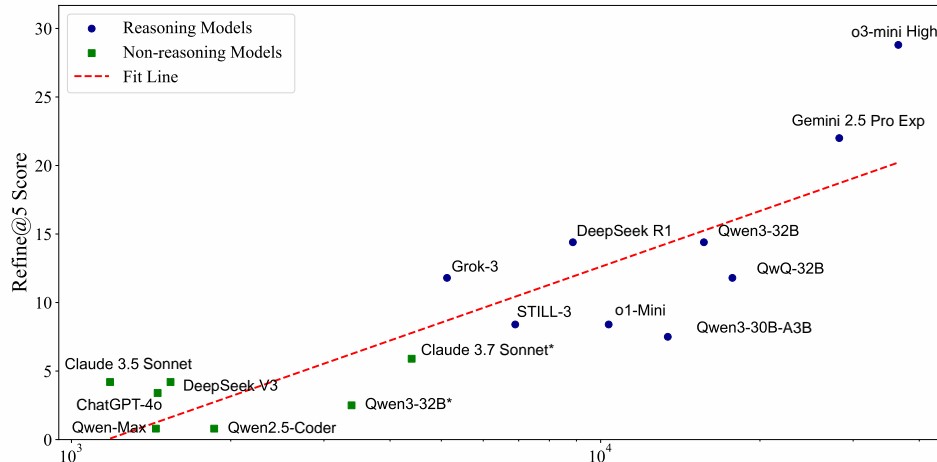

Figure 3: Refine@K scales robustly with increasing output lengths across different models. The output length is measured in tokens.

0.8.5. For **reasoning models**, we include OpenAI o1-mini (via `o1-mini-2024-09-12`), OpenAI o3-mini High (via `o3-mini-2025-01-31-high`), DeepSeek R1 (via `deepseek-reasoner`), Gemini 2.5 Pro Experimental (via `gemini-2.5-pro-exp-03-25`), Grok 3 Mini Thinking Mode [20] (via `grok-3-mini-beta`), QwQ-32B [21], and STILL-3-TOOL-32B [22]. For **non-reasoning models**, we evaluate ChatGPT-4o (via `chatgpt-4o-latest`), Claude 3.5 Sonnet (via `claude-3-5-sonnet-20241022`), DeepSeek V3 0324 (via `deepseek-chat`), and Qwen Max (via `qwen-max-2025-01-25`). Additionally, we assess latest **hybrid reasoning models** including Claude 3.7 Sonnet [23] (non-thinking mode only, via `claude-3-7-sonnet-20250219`), Qwen3-32B (both thinking and non-thinking mode), and Qwen3-30B-A3B [24] (thinking mode only). Due to performance issues encountered while evaluating the thinking mode of Claude 3.7 Sonnet, and our inability to determine if these are caused by the API we used, we are temporarily withholding the results for its thinking mode.

**Evaluation.** For generation hyperparameters, we configure locally-evaluated models with `temperature` 0.6 and `top_p` 0.95. For API-evaluated models, we use their default hyperparameters to better unleash their reasoning capabilities. All generated code is compiled using GNU GCC 14 with the `-std=c++23` flag to ensure maximum compatibility. The program runs on an Intel Xeon Platinum 8160 processor at 2.10 GHz. We use `Refine@5` as the primary evaluation metric. If compilation fails, the error message is returned as feedback. If compilation succeeds, we run the example test cases. Any mismatched outputs are returned alongside the expected outputs. Only code that passes all example tests is further evaluated on hidden test cases, where feedback is limited to the error type (`Wrong Answer`, `Runtime Error`, `Time Limit Exceeded`, `Memory Limit Exceeded`, or `Unknown Error`). Detailed prompts and refinement guidelines are provided in Appendix B.1.

## 4.2 Main Results

In this section, we evaluate the performance of the comparison models on ICPC-Eval and provide a detailed analysis. As provided in Table 3, we have the following observation:

**Execution Feedback Elicits Reflection of Reasoning Models.** As one of our core contributions, we demonstrate that our proposed execution-feedback-based Refine@K metric effectively induces reasoning capabilities in models, enabling more efficient evaluation of test-time scaling. For instance, we found that most of models require more than one turns to generate correct responses. For instance, we found that reasoning models scale effectively as the inference turn budget increases, while non-reasoning models exhibit minimal reflection abilities and scaling potential. We further validate these findings by comparing Refine@K with Pass@K in Section 5.2.

**Different Models Exhibit Expertise in Different Domains.** We find that models vary in domain-specific strengths. For instance, Gemini 2.5 Pro Exp performs well in basic algorithms, data struc-

Table 3: Refine@5 performance of models across various algorithmic domains and full ICPC-Eval test set. Note that a single problem may involve one or more algorithmic domains. The symbol * indicates non-thinking mode for hybrid-reasoning models, while #T represents the average number of correct response turns.

| Models | Domains | | | | | | | | Full | #T |
|---|---|---|---|---|---|---|---|---|---|---|
| | Algo | CG | DP | DS | CT | Math | SA | Str | | |
| **Reasoning Models** | | | | | | | | | | |
| o3-mini High | 26.5 | 17.6 | 21.1 | 33.3 | 23.1 | 29.2 | 16.7 | 50.0 | **28.8** | 1.21 |
| Gemini 2.5 Pro Exp | 20.6 | 5.9 | 13.2 | 30.0 | 11.5 | 22.9 | 0.0 | 37.5 | 22.0 | 1.27 |
| DeepSeek R1 | 11.8 | 0.0 | 10.5 | 23.3 | 11.5 | 8.3 | 0.0 | 25.0 | 14.4 | 2.06 |
| Grok 3 Mini Beta | 17.6 | 0.0 | 7.9 | 10.0 | 7.7 | 10.4 | 0.0 | 25.0 | 11.8 | 1.57 |
| QwQ-32B | 14.7 | 0.0 | 10.5 | 16.7 | 7.7 | 12.5 | 0.0 | 12.5 | 11.8 | 1.57 |
| o1-mini | 8.8 | 0.0 | 5.3 | 10.0 | 7.7 | 12.5 | 0.0 | 25.0 | 8.4 | 1.4 |
| STILL-3-Tool-32B | 8.8 | 0.0 | 7.9 | 6.7 | 0.0 | 10.4 | 0.0 | 25.0 | 8.4 | 1.6 |
| **Hybrid-reasoning Models** | | | | | | | | | | |
| Qwen3-32B | 14.7 | 0.0 | 10.5 | 20.0 | 11.5 | 10.4 | 0.0 | 25.0 | **14.4** | 1.35 |
| Qwen3-30B-A3B | 11.8 | 0.0 | 2.6 | 10.0 | 3.8 | 8.3 | 0.0 | 25.0 | 7.5 | 1.56 |
| Claude 3.7 Sonnet* | 11.8 | 0.0 | 2.6 | 10.0 | 3.8 | 8.3 | 0.0 | 25.0 | 5.9 | 2.2 |
| **Non-reasoning Models** | | | | | | | | | | |
| Claude 3.5 Sonnet | 5.9 | 0.0 | 0.0 | 3.3 | 3.8 | 6.3 | 0.0 | 25.0 | **4.2** | 2.2 |
| Qwen3-32B* | 5.9 | 0.0 | 0.0 | 3.3 | 0.0 | 2.1 | 0.0 | 12.5 | 2.5 | 1.67 |
| DeepSeek V3 | 5.9 | 0.0 | 0.0 | 6.7 | 0.0 | 2.1 | 0.0 | 25.0 | **4.2** | 1.0 |
| ChatGPT-4o | 5.9 | 0.0 | 0.0 | 3.3 | 0.0 | 4.2 | 0.0 | 12.5 | 3.4 | 1.75 |
| Qwen Max | 2.9 | 0.0 | 0.0 | 3.3 | 0.0 | 0.0 | 0.0 | 0.0 | 0.8 | 2.0 |
| Qwen2.5-Coder-32B | 2.9 | 0.0 | 0.0 | 0.0 | 0.0 | 0.0 | 0.0 | 0.0 | 0.8 | 1.0 |

tures, and mathematics, while Grok 3 Mini Beta shows strength only in basic algorithms. Overall, computational geometry and search algorithms are the most challenging areas for LLMs, as they require intricate programming, where minor mistakes can cause failure. Except for o3-mini High and Gemini 2.5 Pro Exp, all other models failed to solve any problems in these two domains.

**Refine@K scales robustly with increasing output lengths.** Figure 3 reveals a positive correlation between Refine@5 scores and the average output length of models. Reasoning models exhibit significantly longer average outputs compared to non-reasoning models, and this trend is mirrored in their Refine@5 scores. Notably, o3-mini High, which has the longest average output length, achieves the highest Refine@5 score. Claude 3.7-nothinking, which produces the longest outputs among non-reasoning models, attains a Refine@K score comparable to that of some reasoning models. This indicates that a longer CoT implies deeper thinking, and refine@K provides an accurate measure of a model's intrinsic reasoning ability.

## 5 Ablation Study

### 5.1 Comparing ICPC-Eval with Other Code Benchmarks

To demonstrate the advantages of ICPC-Eval, we curate several state-of-the-art models across commonly used coding benchmarks [24, 5].

As shown in Table 4, these models all perform worse on ICPC-Eval compared to other benchmarks (*e.g.*, o3-mini High achieves 67.4% on LiveCodeBench vs. 28.8% on ICPC-Eval), highlighting the challenging nature of ICPC-Eval. Moreover, unlike existing benchmarks, where high performance scores limit differentiation, ICPC-Eval produces more varied results, allowing for better discrimination of coding capabilities. For instance, while Grok 3 Mini Beta and o3-mini High achieve comparable accuracy on LiveCodeBench (66.7% vs. 67.4%), their performance diverges substantially on ICPC-Eval (11.8% vs. 28.8%).

Table 4: ICPC-Eval presents a more challenging nature compared to other code benchmarks.

| | ICPC-Eval | LiveCodeBench | CodeElo |
|---|---|---|---|
| | Refine@K | Pass@K | Rating / Percentile |
| o3-mini High | **28.8%** | 67.4% | - |
| Gemini 2.5 Pro Exp | 22.0% | **67.8%** | 2001 / 97.9% |
| DeepSeek R1 | 14.4% | 64.3% | **2029 / 98.1%** |
| Qwen3-32B | 14.4% | - | 1977 / 97.7% |
| Grok 3 Mini Beta | 11.8% | 66.7% | - |
| Claude 3.5 Sonnet | 4.2% | 36.4% | 710 / 24.1% |
| DeepSeek V3 | 4.2% | 27.2% | 1134 / 54.1% |

## 5.2 Comparison of Refine@K and Pass@K

To demonstrate that Refine@K is a more suitable metric than Pass@K for evaluating the code reasoning capabilities of LLMs, we conducted comparative experiments using two model pairs: QwQ-32B vs. Qwen-2.5-Coder-32B (see Figure 4a), and DeepSeek-R1 vs. DeepSeek-V3 0324 (see Figure 4b). Importantly, each pair is derived from the same base model (*i.e.* Qwen-2.5-32B and DeepSeek-V3), which eliminates confounding factors related to differences in pre-training.

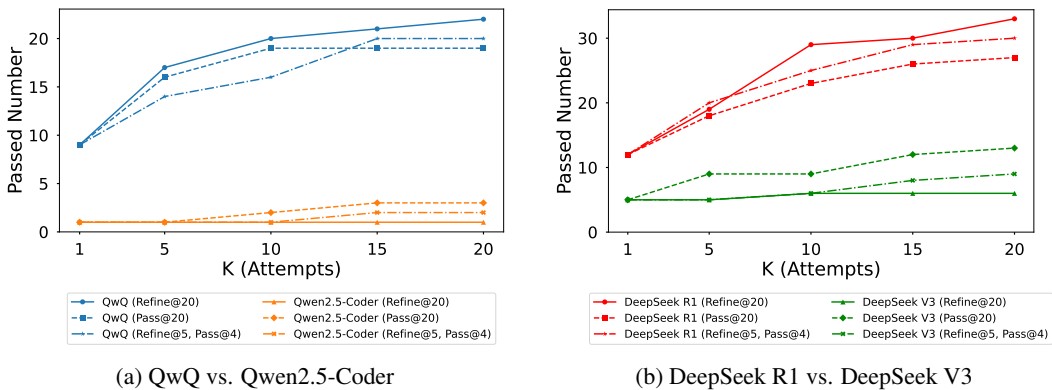

(a) QwQ vs. Qwen2.5-Coder

(b) DeepSeek R1 vs. DeepSeek V3

Figure 4: Comparison of Refine@K and Pass@K methods across different models.

As illustrated in Figure 4, the performance gap between Refine@K and Pass@K widens with an increasing number of attempts for the same model. Specifically, for reasoning models like QwQ-32B and DeepSeek-R1, **Refine@K consistently outperforms Pass@K across all K values**. In contrast, for non-reasoning models such as Qwen-2.5-Coder-32B and DeepSeek-V3, Pass@K is significantly higher than Refine@K, and increasing the number of attempts yields minimal improvement in Refine@K. These findings indicate that reasoning models have the ability to iteratively refine their responses based on previous outputs and feedback, with Refine@K more effectively capturing this behavior than simple rollout. In contrast, non-reasoning models lack these reflective capabilities and may be negatively impacted by prior incorrect responses, resulting in poorer performance compared to simple resampling, as also reported in previous studies [16]. This comparison highlights the fundamental differences in problem-solving abilities between reasoning and non-reasoning models and underscores that Refine@K is a more appropriate metric for assessing the intrinsic capabilities of reasoning LLMs.

As an additional step in this direction, Appendix A reports new evidence that a small reasoning model (Qwen3-1.7B) benefits from Refine@K with increasing $K$, whereas two larger non-reasoning models (Qwen-2.5-Coder-32B, DeepSeek-V3) show limited gainsupporting the view that parameter scaling and test-time scaling follow distinct dynamics.

## 6 Conclusion, Limitation, and Future Work

In this work, we introduce **ICPC-Eval**, a challenging benchmark consisting of 118 carefully selected competitive programming problems from recent ICPC contests, together with a robust local

evaluation pipeline and the test-time scaling metric **Refine@K**. Our results show that, despite steady progress, state-of-the-art models still exhibit a substantial gap to top human teams on ICPC-Eval. We further observe that Refine@K more faithfully captures feedback-driven, iterative problem solving than Pass@K, particularly for reasoning models, and scales robustly with output length.

**Limitations.** While ICPC-Eval aims to emulate realistic contest conditions, several limitations remain. (1) *Scope*: the current release focuses on 11 recent contests and primarily features C++ problems from Asia, Europe, and North America; expanding geographic and temporal coverage will improve representativeness. (2) *Language coverage*: we currently evaluate C++ to align with ICPC practice; extending to additional languages (e.g., Python, Java) is an important next step. (3) *Task modality*: problems that fundamentally rely on images or interactive protocols are excluded; evaluating multimodal and interactive tasks is left for future work.

**Future Work.** We will periodically refresh ICPC-Eval with new contests to reduce contamination risk and broaden the distribution of tasks and regions; add multilingual evaluation beyond C++; and extend to multimodal and interactive settings. Beyond core LLMs, the framework naturally supports tool-augmented agents (e.g., debuggers and profilers).

These findings highlight the rigor and importance of ICPC-Eval as a benchmark for advancing the study of reasoning in large language model-based programming.

## Acknowledgments

This work was partially supported by National Natural Science Foundation of China under Grant No. 92470205 and 62222215, Beijing Natural Science Foundation under Grant No. L233008 and Beijing Municipal Science and Technology Project under Grant No. Z231100010323009. Xin Zhao is the corresponding author.

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

# A    Additional Experiments: Refine@K vs. Pass@K under Varying Model Types

Table 5: Problems solved versus $K$ for a small reasoning model and two larger non-reasoning models.

| Model | Setting | $K=1$ | $K=5$ | $K=10$ | $K=15$ | $K=20$ |
|---|---|---|---|---|---|---|
| Qwen3-1.7B | Refine@20 | 1 | 2 | 2 | 4 | 4 |
| Qwen3-1.7B | Pass@20 | 1 | 2 | 3 | 3 | 3 |
| Qwen3-1.7B | Refine@5, Pass@4 | 1 | 2 | 2 | 3 | 3 |
| Qwen-2.5-Coder-32B | Refine@20 | 1 | 1 | 1 | 1 | 1 |
| Qwen-2.5-Coder-32B | Pass@20 | 1 | 1 | 2 | 3 | 3 |
| Qwen-2.5-Coder-32B | Refine@5, Pass@4 | 1 | 1 | 1 | 2 | 2 |
| DeepSeek-V3 | Refine@20 | 5 | 5 | 6 | 6 | 6 |
| DeepSeek-V3 | Pass@20 | 5 | 9 | 9 | 12 | 13 |
| DeepSeek-V3 | Refine@5, Pass@4 | 5 | 5 | 6 | 8 | 9 |

We supplement the main results with a controlled study contrasting a small reasoning model (Qwen3-1.7B) with two larger non-reasoning models (Qwen-2.5-Coder-32B and DeepSeek-V3). The table below reports the number of problems solved as $K$ increases under three settings: Refine@20, Pass@20, and a matched budget comparison (Refine@5 vs. Pass@4). The results show a clear upward trend for the reasoning model under Refine@K, whereas the two non-reasoning models exhibit minimal or inconsistent improvements under Refine@K despite stronger capacitysupporting that Refine@K captures a distinct feedback-driven reasoning capability separate from raw model size.

# B    Prompts

## B.1    Prompt Used for ICPC-Eval

Initial generation:

You are a coding expert. Given a competition-level coding problem, you need to write a C++ program (C++23) to solve it. Please consider the efficiency and time complexity of the algorithm to meet the time limit requirements of the problem. You may start by outlining your thought process.
In the end, YOU MUST provide the complete code in a code block enclosed with "' "'.
In the end, YOU MUST provide the complete code in a code block enclosed with "' "'.
In the end, YOU MUST provide the complete code in a code block enclosed with "' "'.
Problem: {title}
Time limit: {time_limit_ms}ms
Memory limit: {memory_limit_mb}MB
[Description]
{description}
[Input]
{input}
[Output]
{output}
[Sample Input i]
{sample_input_i}
[Sample Output i]
{sample_output_i}
...
[Note]
{note}

Refinement (Fail on Sample Test Cases):

...
The code you generated encountered an error when tested locally: {correct_info}. {Suggestion}. Please modify your code. You should analyze the reasons for the error. You may start by outlining your thought process.
In the end, YOU MUST provide the complete code in a code block enclosed with "' "'.
In the end, YOU MUST provide the complete code in a code block enclosed with "' "'.
In the end, YOU MUST provide the complete code in a code block enclosed with "' "'.

Refinement(Fail on Final Test Cases):

...
The code you generated encountered an error after submitting to the Contest Judge: {correct_info}. {Suggestion}. Please modify your code. You should analyze the reasons for the error. You may start by outlining your thought process.
In the end, YOU MUST provide the complete code in a code block enclosed with "' "'.
In the end, YOU MUST provide the complete code in a code block enclosed with "' "'.
In the end, YOU MUST provide the complete code in a code block enclosed with "' "'.

## B.2 Prompt Used for Annotating Algorithmic Domains

**[Category 1]: Algorithm Basics**
Enumeration, Simulation, Recursion & Divide and Conquer, Greedy, Sorting, Binary Search, Doubling, Construction
**[Category 2]: Search**
DFS, BFS, Bidirectional Search, Heuristic Search, A*, Iterative Deepening Search, IDA*, Backtracking, Dancing Links, Alpha-Beta Pruning, Other Search Methods
**[Category 3]: Dynamic Programming (DP)**
Introduction to Dynamic Programming, Basic Dynamic Programming, Memoization, Knapsack DP, Interval DP, DP on DAGs, Tree DP, Bitmask DP, Digit DP, Plug DP, Counting DP, Dynamic DP, Probability DP, DP Optimization, Other DP Methods
**[Category 4]: Advanced String Algorithms**
String Matching, String Hashing, Trie, Prefix Function and KMP Algorithm, BoyerMoore Algorithm, Z-Function (Extended KMP), Automaton, AhoCorasick Automaton, Suffix Array (SA), Suffix Automaton (SAM), Suffix Balanced Tree, Generalized Suffix Automaton, Suffix Tree, Manacher, Palindrome Tree, Sequence Automaton, Minimal Representation, Lyndon Decomposition, MainLorentz Algorithm
**[Category 5]: Mathematics**
Number Systems, Bit Manipulation, Binary Set Operations, Balanced Ternary, High-Precision Arithmetic, Fast Exponentiation, Permutations and Combinations, Radians and Coordinate Systems, Complex Numbers, Number Theory, Polynomials and Generating Functions, Combinatorics, Linear Algebra, Linear Programming, Abstract Algebra, Probability Theory, Game Theory, Numerical Algorithms, FourierMotzkin Elimination, Order Theory, Young Tableaux, Matroid, BerlekampMassey Algorithm
**[Category 6]: Advanced Data Structures**
Stack, Queue, Linked List, Hash Table, Disjoint Set Union, Heap, Block Data Structures, Monotonic Stack, Monotonic Queue, Sparse Table (ST), Binary Indexed Tree (Fenwick Tree), Segment Tree, Partition Tree, Binary Search Tree & Balanced Tree, Skip List, Persistent Data Structures, Tree of Trees, K-D Tree, Dynamic Tree, Decomposition Tree, PQ Tree, Finger Tree, Huffman Tree, Loser Tree
**[Category 7]: Graph Theory**
Graph Representation, DFS (Graph Theory), BFS (Graph Theory), Tree Problems, Matrix-Tree Theorem, Directed Acyclic Graphs, Topological Sort, Minimum Spanning Tree, Steiner Tree, Minimum Arborescence, Minimum Diameter Spanning Tree, Shortest Path, Vertex Splitting, Difference Constraints, k-Shortest Paths, Congruent Shortest Path, Connectivity, Cycle Counting Problems, 2-SAT, Eulerian Graph, Hamiltonian Graph, Bipartite Graph, Minimum Cycle
**[Category 8]: Computational Geometry**
2D Computational Geometry Basics, 3D Computational Geometry Basics, Distance, Pick's Theorem, Triangulation, Convex Hull, Sweep Line, Rotating Calipers, Half-Plane Intersection, Closest Pair of Points, Randomized Incremental Algorithm, Inversion Transformation, Miscellaneous Computational Geometry
The above are the 8 categories of algorithm competition problems and their subcategories. Next, I will provide you with an algorithm problem and its correct code solution. Please read them and determine which of the above 8 categories the problem belongs to. Each problem may belong to multiple categories. If a problem involves any subcategory under a category, it is considered to belong to that category. First, introduce the core algorithms involved in the problem, then output the categories the problem belongs to in Python list format, e.g., ["Category1EnglishName", "Category2EnglishName", ...], using the names of the categories. Then, explain each category inclusion one by one.

## B.3 Prompt Used for Synthesizing Test Cases

Random case Generator:

> You are a programming contest expert. Given a competitive programming problem and it's standard solution code, you need to write a C++ program(C++11) to generate random test input data for the problem. Please ensure that the generated test data satisfies all constraints in the problem description. Your C++ program should generate a set of valid test input data when executed, which should test the correctness and efficiency of solutions. The range of generated random data should be consistent with the requirements of the problem, do not use small range for simplicity. Your program must use the system's default time as the random seed and output only the test input data (without any extra prompts or commentary). In the end, YOU MUST provide the complete C++ code in a code block enclosed with " "'!!! YOU MUST provide the complete C++ code in a code block enclosed with " "'!!! YOU MUST provide the complete C++ code in a code block enclosed with " "'!!!

Corner case Generator:

> You are a programming contest expert. Given a competitive programming problem and its standard solution code, you need to write a C++ (C++11) program that generates diverse random test input data for the problem. Unlike standard generators, your program must randomly decide at runtime which type of test input to produce, choosing from multiple types that include edge cases, boundary extreme values, and specially structured cases. You must ensure that the input data generated after each run of this generator and its output data is greatly different and diverse. The generated data must satisfy all constraints detailed in the problem description and cover the full range of allowed values, ensuring that any submitted solution is thoroughly tested for both correctness and efficiency. Your program must use the system's default time as the random seed and output only the test input data (without any extra prompts or commentary). In the end, YOU MUST provide the complete C++ code in a code block enclosed with " "'!!! YOU MUST provide the complete C++ code in a code block enclosed with " "'!!! YOU MUST provide the complete C++ code in a code block enclosed with " "'!!!

# C   Dataset Details

## C.1   Selected ICPC Contests

Table 6: Selected ICPC Contests.

| Contest | Category |
|---|---|
| The 2023 ICPC World Finals | World Final |
| The 2024 ICPC Asia East Continent Final Contest | Continent Final |
| The 2024 ICPC North America Championship | Continent Final |
| The 2024 ICPC Asia Chengdu Regional Contest | Regional |
| The 2024 ICPC Asia Hangzhou Regional Contest | Regional |
| The 2024 ICPC Asia Hong Kong Regional Contest | Regional |
| The 2024 ICPC Asia Nanjing Regional Contest | Regional |
| The 2024 ICPC Asia Shanghai Regional Contest | Regional |
| The 2024 ICPC Asia Shenyang Regional Contest | Regional |
| The 2024 ICPC Northwestern Europe Regional Contest | Regional |
| The 2024 ICPC Central Europe Regional Contest | Regional |

