# OpenReview forum: "ICPC-Eval: Probing the Frontiers of LLM Reasoning with Competitive Programming Contests"
_NeurIPS.cc/2025/Datasets_and_Benchmarks_Track — NeurIPS 2025 Datasets and Benchmarks Track poster_

### Official Review · Reviewer_quup · 2025-06-09

**Rating:** 4
**Confidence:** 4

**Summary:**

The authors introduce ICPC-Eval, a new benchmark for evaluating large language models on challenging competitive programming problems sourced from the ICPC. The paper's primary contributions are: 1) the introduction of a challenging new dataset of realistic ICPC problems, 2) a novel methodology for generating test data using LLMs, and 3) the proposal of a new evaluation metric, Refine@k.

**Dataset Code Accessibility:**

Yes

**Ethical Considerations:**

No, there are no or only very minor ethics concerns

**Final Justification:**

I have updated my rating. The experiments presented effectively highlight the importance of the proposed refine@k in the reasoning model, addressing key concerns raised earlier.

**Limitations Weaknesses:**

The paper does not sufficiently address the limitations of the refine@k metric. While pass@k is naturally parallelizable and thus scalable, refine@k introduces sequential dependencies that make it difficult to implement efficiently in parallel settings. This restricts its practical applicability in large-scale deployments and should be discussed.

Moreover, the key claim—that reasoning models benefit from refine@k whereas non-reasoning models do not—is not fully convincing given the current experimental setup. In particular, the reasoning model used in the study is significantly stronger in general code understanding than the non-reasoning baseline. This raises the possibility that the observed improvements stem more from model capacity than from reasoning ability.

To better support the central hypothesis, I suggest including additional experiments: for example, evaluating whether a weaker model with reasoning capabilities (e.g., Qwen3-1.7B) benefits from refine@k, while a stronger non-reasoning model (e.g., Claude 3.5 Sonnet) does not. Such evidence would more clearly isolate the effect of reasoning over raw model capacity and strengthen the paper’s core argument.

**Strengths Contributions:**

The paper effectively distinguishes its benchmark by highlighting the high difficulty of ICPC problems. The manuscript itself is well-organized and easy to understand. Furthermore, the domain-wise performance analysis provides a valuable and granular method for evaluating LLM capabilities.

---

> ### Author Rebuttal · Authors · 2025-07-31
>
> **Q1: The paper does not sufficiently address the limitations of the refine@k metric. While pass@k is naturally parallelizable and thus scalable, refine@k introduces sequential dependencies that make it difficult to implement efficiently in parallel settings.**
>
> **A1:** Since the evaluation of each of the 118 problems in the ICPC‑Eval benchmark is conducted independently, we can employ two parallelization strategies: problem-level parallelism and rollout-level parallelism. For a single problem instance, Refine@K cannot execute multiple rollouts in parallel in the same way as Pass@K can, but it still allows concurrent evaluation across multiple problems. Therefore, it does not compromise the efficiency of large-scale evaluation.
>
> **Q2: The key claim—that reasoning models benefit from refine@k whereas non-reasoning models do not—is not fully convincing given the current experimental setup... the observed improvements stem more from model capacity than from reasoning ability.**
>
> **A2:** Thank you for your question. Our work does not aim to propose a superior method for improving model performance. Instead, we introduce Refine@K as a better metric compared to Pass@K for evaluating model's reasoning capabilities and test-time scaling. We have conducted an additional experiment and compared the results of a small reasoning model **Qwen3-1.7B** with two larger non‑reasoning models as suggested:
>
> **Qwen3-1.7B**
> | Passed Number        |  K=1  |  K=5  | K=10  | K=15  | K=20  |
> | :------------------- | :---: | :---: | :---: | :---: | :---: |
> | **Refine@20**        |   1   |   2   |   2   |   4   |   4   |
> | **Pass@20**          |   1   |   2   |   3   |   3   |   3   |
> | **Refine@5, Pass@4** |   1   |   2   |   2   |   3   |   3   |
>
> **Qwen-2.5-Coder-32B:**
> | Passed Number        |  K=1  |  K=5  | K=10  | K=15  | K=20  |
> | :------------------- | :---: | :---: | :---: | :---: | :---: |
> | **Refine@20**        |   1   |   1   |   1   |   1   |   1   |
> | **Pass@20**          |   1   |   1   |   2   |   3   |   3   |
> | **Refine@5, Pass@4** |   1   |   1   |   1   |   2   |   2   |
>
> **DeepSeek-V3:**
> | Passed Number        |  K=1  |  K=5  | K=10  | K=15  | K=20  |
> | :------------------- | :---: | :---: | :---: | :---: | :---: |
> | **Refine@20**        |   5   |   5   |   6   |   6   |   6   |
> | **Pass@20**          |   5   |   9   |   9   |  12   |  13   |
> | **Refine@5, Pass@4** |   5   |   5   |   6   |   8   |   9   |
>
> As the results show, while the overall performance is limited by Qwen3-1.7B's parameter scale, `Refine@20` (4 solved) still outperforms `Pass@20` (3 solved). More importantly, the `Refine@K` scores for the small reasoning model exhibits **a clear upward trend as K increases**, but the `Refine@K` scores for the two large non-reasoning models **barely improve with more attempts**. This new evidence strongly supports our hypothesis that `Refine@K` captures a distinct "reasoning" or "reflective" capability that is separate from general model capacity. Although both parameter scaling (increasing model size) and test‑time scaling (via refinement) can improve model performance, they follow two distinct scaling laws, and thus the model exhibits distinct different behavioral characteristics under different approach. `Refine@K` provides a more precise evaluation of a model’s test-time scaling capability.
>
> We may incorporate these new experimental results and clarifications into the later version of the paper. We hope these additions can answer your questions.

---

> > ### Comment · Reviewer_quup · 2025-08-02
> >
> > Thank you for the excellent work and thorough explanation. I decided to increase my rating.

---

> > > ### Author Response · Authors · 2025-08-05
> > >
> > > Thank you so much for increasing the rating. We would greatly appreciate the opportunity to engage in discussions with you. The discussion will be added into our paper. Your insightful comments have been extremely valuable in helping us strengthen our work and clarify our contributions. We are confident that by incorporating this discussion and the new experimental results, the final version of the paper will be significantly improved. Thank you once again for your constructive feedback and support.

---

### Official Review · Reviewer_fcbD · 2025-06-26

**Rating:** 5
**Confidence:** 3

**Summary:**

This paper introduces ICPC-Eval, a competitive programming benchmark designed to evaluate the advanced reasoning capabilities of large language models (LLMs) in a realistic competition setting. The benchmark comprises 118 problems carefully curated from 11 recent International Collegiate Programming Contest (ICPC) events, ensuring a challenging problem set with a difficulty distribution consistent with actual contests. Key contributions include a robust test case generation method leveraging LLMs to create comprehensive local test suites, enabling efficient and accurate offline evaluation, and a novel evaluation metric, Refine@K, which assesses an LLM's ability to iteratively refine its solutions based on execution feedback over multiple attempts. Baseline evaluations on 15 state-of-the-art LLMs indicate that even top-tier reasoning models like DeepSeek-R1 often rely on multi-turn code feedback to unlock their in-context reasoning potential and still lag behind human performance, highlighting the difficulty of the benchmark and significant opportunities for future research. The benchmark and tools are publicly released.

**Dataset Code Accessibility:**

Yes

**Dataset Code Comments:**

The dataset has been released.

**Ethical Considerations:**

No, there are no or only very minor ethics concerns

**Final Justification:**

My concerns are addressed so I maintain the original positive rating.

**Limitations Weaknesses:**

- **Limited Problem Scope**: While the paper aims for high difficulty, it explicitly excludes problems that involve essential non-textual images or interactive elements. This design choice, though pragmatic for current text-based LLMs, limits the benchmark's ability to assess multimodal reasoning, which is increasingly relevant for complex real-world problems. Extending the benchmark to include such problems could be a valuable future direction.

- **Manual Tagging with AI Assistance**: The algorithmic tagging of problems was performed manually with assistance from a Gemini model, which involved reviewing AI-generated explanations. While the manual review step adds a layer of quality control, relying on an LLM for initial classification suggestions could introduce subtle biases or inaccuracies if not meticulously checked, particularly for complex algorithmic domains.

- **Generalizability of Refine@K to Non-Programming Tasks**: While Refine@K is well-suited for code generation with execution feedback, its direct applicability to other types of reasoning tasks that do not involve explicit, objective, and iterative feedback mechanisms might be limited. The paper primarily focuses on its benefits within the competitive programming context.

**Strengths Contributions:**

- **Challenging and Realistic Problem Set**: ICPC-Eval addresses the limitations of existing benchmarks by providing a collection of high-difficulty problems from actual ICPC contests. This ensures a rigorous test of advanced reasoning that is less susceptible to data contamination and memorization compared to simpler benchmarks. The problem curation process, including filtering for non-textual images and interactive elements, contributes to its robustness.

- **Robust Local Evaluation Infrastructure**: The paper introduces a novel methodology for generating comprehensive local test cases using LLMs, which is a practical solution to the common issue of inaccessible private test cases in competitive programming platforms. The validation process, which involves checking against known incorrect programs, contributes to ensuring zero false positives, thereby enhancing the reliability of the evaluation.

- **Comprehensive Baseline Evaluation and Analysis**: The paper provides an extensive evaluation of 15 state-of-the-art LLMs, offering detailed insights into their performance across various algorithmic domains. The results clearly show the substantial gap between LLM performance and human medalists, indicating the challenging nature of the benchmark and areas for future research. The observation that reasoning models benefit from multi-turn feedback and longer outputs is a valuable finding.

---

> ### Author Rebuttal · Authors · 2025-07-31
>
> **Q1: Limited Problem Scope: While the paper aims for high difficulty, it explicitly excludes problems that involve essential non-textual images or interactive elements.**
>
> **A1:** Thank you for your comment. To accommodate most reasoning models, we converted a subset of image-based problems into pure text using OCR and manual curation. Only a small number of problems that fundamentally rely on image understanding or interactive execution were excluded. The remaining set is already sufficiently challenging to assess a model’s programming competition ability. We agree that problems requiring visual comprehension or interactive behavior are intriguing, and we view them as valuable directions for future work.
>
> **Q2: Manual Tagging with AI Assistance: ...relying on an LLM for initial classification suggestions could introduce subtle biases or inaccuracies if not meticulously checked.**
>
> **A2:** Thank you for raising this concern. First, we’d like to clarify that the tags are used solely for statistical categorization of problems and do not play any role in the actual evaluation process, so they do not affect the credibility of the final benchmark results. Second, while we used LLMs to assist with initial suggestions, all final tags were verified and curated by human experts to ensure accuracy and consistency.
>
> **Q3: Generalizability of Refine@K to Non-Programming Tasks: ...its direct applicability to other types of reasoning tasks that do not involve explicit, objective, and iterative feedback mechanisms might be limited.**
>
> **A3:** This is an excellent point. Refine@K is initially tailored for programming, where iterative feedback is explicit and objective. Rather than viewing this as a limitation, we consider it a core strength. The metric closely mirrors the real-world iterative refinement process that both humans and advanced models rely on. More broadly, Refine@K offers a concrete instantiation of modeling feedback-driven iteration, and serve as a proxy to evaluate the test time scaling potential of a model.

---

> > ### Comment · Reviewer_fcbD · 2025-08-04
> >
> > Thanks for your response. My concerns have been addressed. I will maintain my positive rating.

---

> > > ### Author Response · Authors · 2025-08-05
> > >
> > > Thank you for your valuable feedback and for maintaining your positive rating. We are grateful for your recognition of our work. Your insightful comments have been very helpful. We sincerely appreciate your support.

---

### Official Review · Reviewer_BqCc · 2025-06-30

**Rating:** 4
**Confidence:** 2

**Summary:**

ICPC-Eval introduces a challenging benchmark of 118 ICPC competition problems to evaluate LLM reasoning, featuring a local evaluation pipeline with LLM-generated test cases and the novel Refine@K metric that measures iterative code refinement using execution feedback, revealing significant gaps where top models lag human medalists while reasoning models outperform non-reasoning counterparts in feedback utilization.

**Additional Feedback:**

1. Incorporate partial credit scoring for passing test subsets to mirror ICPC grading.
2. Add human baseline turn counts to contextualize Refine@K efficiency.
3. Evaluate tool-augmented agents (e.g., debuggers) to bridge human-model gaps.

**Dataset Code Accessibility:**

Yes

**Dataset Code Comments:**

Details for reproducing are contained in the paper. The code is well-documented, with datasets publicly accessible.

**Ethical Considerations:**

No, there are no or only very minor ethics concerns

**Final Justification:**

Since I am not familiar with this research, I choose to keep the score.

**Limitations Weaknesses:**

1. Exclusive C++ focus limits applicability to multilingual code generation models.
2. Pipeline bias risk: Heavy Gemini dependency for labeling/test generation may propagate model-specific artifacts.
3. Only 3 continents covered potentially skews algorithmic diversity.

**Strengths Contributions:**

1. 118 authentic ICPC problems ensure unprecedented difficulty and realism, exposing critical reasoning gaps in SOTA models.
2. LLM-synthesized test generators enable scalable offline assessment with zero false positives via adversarial validation.
3. Measures iterative refinement capability using execution feedback, revealing reasoning models' advantage in multi-turn repair.

---

> ### Author Rebuttal · Authors · 2025-07-31
>
> ### **Weaknesses**
>
> **Q1: The exclusive focus on C++ limits applicability to multilingual code generation models.**
>
> **A1:** Thank you for raising this point. While most existing code generation benchmarks focus exclusively on Python—which is suitable for many real-world applications—this does not reflect the needs of high-performance scenarios such as algorithm competitions. C++ remains the dominant language in ICPC and similar contests, so our current implementation focuses on C++ to align with the majority of participants. However, we fully agree that supporting multiple languages (e.g., Python, Java) is an important future direction, and plan to support other languages in future updates.
>
> **Q2: Pipeline bias risk due to dependency on Gemini for labeling and test generation.**
>
> **A2:** We used the Gemini model in the two steps below:
>
> 1. **Algorithm tagging:** Gemini is used only to provide initial classification suggestions. The final labels are manually verified and corrected by human experts based on problem statements and solutions, ensuring that no model-specific heuristics affect the annotations.
>
> 2. **Test case generation:** Gemini is employed to synthesize **input generator code**, not the test cases themselves. These generators are then validated using accepted and incorrect human-written solutions collected from online judges. This adversarial validation ensures that test cases are reliable and not biased by the LLM's output.
>
> **Q3: The benchmark covers only three continents, which may limit algorithmic diversity.**
>
> **A3:** The current dataset already exhibits considerable geographical diversity, comprising Regional contests from Asia, Europe, and North America, in addition to the World Finals. These three continents host the largest number of ICPC participants, and over 80% of teams in the ICPC World Finals also originate from these regions, thereby reflecting the actual geographic representation and diversity in ICPC competition.
>
> ### **Additional Feedback**
>
> **Q4: Consider incorporating partial credit for solutions passing a subset of test cases.**
>
> **A4:** To maintain fidelity with official ICPC scoring rules, our evaluation follows a strict binary criterion: a problem is considered solved only if all test cases are passed. ICPC contests do not assign partial credit or distinguish among subsets of test cases. Therefore, ICPC-Eval retains this all-or-nothing evaluation paradigm to reflect the authentic difficulty and high stakes of ICPC competitions.
>
> **Q5: Include human baseline turn counts to contextualize Refine@K.**
>
> **A5:** Thank you for the suggestion. Analyzing the official contest data from The 2024 ICPC Asia Chengdu Regional Contest, we find that ICPC teams submit an average of **1.95 attempts** per solved problem. This figure significantly exceeds the average `#T` values for all models in **Table 3**, underscoring that humans possess strong reasoning abilities and top human teams rely heavily on feedback-driven refinement. We will subsequently collect human baseline results for the remaining contests and incorporate them into the paper.
>
> **Q6: Evaluate tool-augmented agents (e.g., debuggers) to close the human-model gap.**
>
> **A6:** We agree that evaluating tool-augmented agents is a valuable direction for future research. Our current benchmark focuses on the core in-context reasoning abilities of LLMs in isolation. The ICPC-Eval framework is designed to be extensible and can support future evaluations that integrate external tools such as debuggers or profilers. We view this as a natural and promising next step.

---

> > ### Comment · Reviewer_BqCc · 2025-08-05
> >
> > Thank you for your rebuttal. I have no further questions and since I am not familiar with this research, I choose to keep the score.

---

> > > ### Author Response · Authors · 2025-08-05
> > >
> > > Thank you for your detailed feedback. We sincerely appreciate the time and effort you have dedicated to reviewing our work. We are glad our responses addressed your concern. Wish you a pleasant day.

---

### Official Review · Reviewer_eS9D · 2025-07-03

**Rating:** 6
**Confidence:** 5

**Summary:**

This paper presents ICPC-Eval, a high-difficulty code reasoning benchmark that contains 118 curated problems from recent ICPC contests across multiple regions and difficulty levels. The authors propose a comprehensive evaluation pipeline that includes local test case generation, special judge support, and a novel test-time scaling metric Refine@K. They then benchmark 15 leading LLMs under this framework to assess iterative reasoning capabilities with execution feedback. The paper experiments show a significant performance gap between top LLMs and human medalists. This demonstrates that current models still fall short in realistic, high-stakes competitive programming environments that require multi-step algorithmic reasoning.

**Dataset Code Accessibility:**

Yes

**Ethical Considerations:**

No, there are no or only very minor ethics concerns

**Final Justification:**

I think this is a well-qualified work and would interest the community. I will keep the high score.

**Limitations Weaknesses:**

I see the following minor issues with the setup and evaluation:

i. The problem selection is somewhat limited to ICPC contests from specific years and regions. This may not fully reflect the diversity of competitive programming challenges.

ii. How to scale the benchmark and keep judging consistency for more diverse and/or open-ended programming problems should be better presented (especially considering the reliance on special judges and execution environments).

iii. Although this method focuses on test-time refinement and feedback-driven reasoning, the analysis does not clearly provide sufficient insights into the iterative improvement behavior across different models, e.g., a detailed study on how performance evolves with Refine@K under varying initial quality levels.

**Strengths Contributions:**

The proposed benchmark has been meticulously constructed, featuring authentic problem distributions and test environments of ICPC-level competitive programming contests. It includes various real-world problems across multiple algorithmic domains and contest tiers, and it offers a fine-grained and comprehensive evaluation framework with local test case generation.

Overall this paper is well-written and easy to follow. The figures clearly demonstrate performance gaps between LLMs and human medalists, and visualize the impact of output length and iteration on solution quality. Further, the code is well structured and documented.

The experimental results underscore the benchmark's challenges for current reasoning-oriented LLMs and yield valuable insights for future research.

---

> ### Author Rebuttal · Authors · 2025-07-31
>
> **Q1: The problem selection is somewhat limited to ICPC contests from specific years and regions. This may not fully reflect the diversity of competitive programming challenges.**
>
> **A1:**
>
> Thank you for your insightful comment. ICPC-Eval is curated from 11 ICPC contests held in recent years. Our selection follows two primary criteria: (1) **minimal contamination**, to evaluate the latest models on up-to-date and unseen problems, reducing the likelihood that these problems were included in LLM training corpora; (2) **representativeness**, ensuring that the selected contests reflect the typical distribution of problem types and difficulty levels found in high-level ICPC competitions. While this curated selection already provides a rigorous and contemporary benchmark, we plan to update ICPC-Eval periodically with the latest ICPC problems, making it an evolving resource for the community.
>
> **Q2: How to scale the benchmark and keep judging consistency for more diverse and/or open-ended programming problems should be better presented.**
>
> **A2:**
> 1. **Scalability:** To efficiently scale the benchmark, we use LLMs to synthesize test case generators to sample random and corner cases. This process reduces manual effort to produce extensive test coverage, enabling efficient expansion to new problems.
>
> 2. **Consistency & Diversity:** While verifying open-ended tasks like web development remains a known challenge, it's beyond the primary scope of this benchmark. For the vast majority of programming competition problems, our framework ensures consistent and fair evaluation through synthesized test case generators and support for special judges.
>
>
> **Q3: The analysis does not clearly provide sufficient insights into the iterative improvement behavior across different models, e.g., a detailed study on how performance evolves with Refine@K under varying initial quality levels.**
>
> **A3:**
>
> We analyze the patterns in how different models handle errors of varying difficulty. We find that reasoning models exhibit a bug-fixing ability distribution that more closely resembles that of humans compared to non-reasoning models. While most models can fix a simple compile error, only reasoning models consistently tackle issues that require abstract thought, such as redesigning an algorithm to fix a TLE or hypothesizing about hidden test cases.
>
> | **Initial Error**                  | **Possible Causes**                            | **Bug Fix Success Rate**                                                                         |
> | ---------------------------------- | ---------------------------------------------- | ------------------------------------------------------------------------------------------------ |
> | **Compile Error**                  | Minor syntax or typographical mistakes.        | **High.**                                                                 |
> | **Wrong Answer (on Samples)**      | Minor bugs or flawed algorithm design.         | **Moderate.** Reasoning models show clear strength in logical debugging.                         |
> | **Runtime Error**                  | Poor coding practices or unhandled edge cases. | **Low.** The performance gap between reasoning and non-reasoning models becomes more pronounced. |
> | **Time / Memory Limit Exceeded**   | Inefficient algorithms or data structures.     | **Very low.**                               |
> | **Wrong Answer (on Hidden Tests)** | Fundamental flaws in algorithm design.         | **Extremely low.** Models struggle without feedback on hidden cases.                             |
>
> This demonstrates that `Refine@K` serves as a reliable indicator for evaluating the test-time scaling behavior of reasoning models.

---

> > ### Comment · Reviewer_eS9D · 2025-08-05
> >
> > Thank you for your reply. I think this is a well-qualified work and would interest the community.

---

> > > ### Author Response · Authors · 2025-08-07
> > >
> > > Thank you for your recognition. We sincerely appreciate the time and effort you have dedicated to reviewing our work. We are glad our responses addressed your concern. Wish you a pleasant day.

---

### Decision · Program_Chairs · 2025-09-18

**Decision:**

Accept (poster)

**Comment:**

This paper presents ICPC-Eval, a challenging code-reasoning benchmark of problems selected from the International Collegiate Programming Contest - a prestigious college competitive programming competition. This benchmark includes test cases generated by an LLM and uses the metric Refine@K (instead of Pass@K) giving the models K shots to refine their solutions against execution feedback.  Evaluations on several State of the Art reasoning benchmarks demonstrate the challenge of this benchmark for current models and the gap to human medalists.

# Strengths:
* Many reviewers agreed that the problem distribution was both challenging and an accurate reflection of real world competitive programming.
* Many reviewers also looked favourably upon the LLM-based test case generation pipeline, in allowing users to evaluate the eval locally.
* Reviewer eS9D highlighted both the quality of the writing (also noted by Reviewer quup), and also the performance gap to human medallists. Reviewer BqCc, looked favourably on the Refine@K metric, in particular for its ability to assess “iterative refinement capabilities”.

# Weaknesses:
* Some questions were raised about the limited scope of the puzzles, whether by virtue of being limited to the ICPC (Reviewer eS9D) or by being overwhelmingly C++ or from only 3 continents (Reviewer BqCc) or being text only (fcbD). The authors noted these in turn, pointing out the focus of the eval on ICPC (predominantly in C++, in Europe, Asia, North America), their efforts to render image questions in text, and the scope of the eval.
* Reviewer quup raise serious concerns about the Refine@k metric, questioning its value and the given how unparallelizable it is. They proposed some experiments, which were duly followed up on by the authors, presenting the phenomena that was expected, and causing Reviewer quup to raise their score.

# AC Note:
Following the unanimous recommendation of the reviewers, I am recommending this paper for acceptance at the conference. Alongside the positive comments made about the challenge of the models, the gap to human performance, the ‘realism’ of the tasks, and the ease of ‘local evaluation’, I was particularly heartened to see the engagement of the authors to fully address the concerns Reviewer quup had about the Refine@K metric.